# Cross-channel Communication Networks

**Jianwei Yang**[1]    **Zhile Ren**[1]    **Chuang Gan**[3]    **Hongyuan Zhu**[4]    **Devi Parikh**[1,2]
[1]Georgia Institute of Technology, [2]Facebook AI Research,
[3] MIT-IBM Watson AI Lab, [4]Institute for Infocomm Research, A*Star, Singapore

## Abstract

Convolutional neural networks (CNNs) process input data by feed-forwarding responses to subsequent layers. While a lot of progress has been made by making networks deeper, filters at each layer independently generate responses given the input and do not communicate with each other. In this paper, we introduce a novel network unit called Cross-channel Communication (C3) block, a simple yet effective module to encourage the communication across filters within the same layer. The C3 block enables filters to exchange information through a micro neural network, which consists of a feature encoder, a message passer, and a feature decoder, before sending the information to the next layer. With C3 block, each channel response is modulated by accounting for the responses at other channels. Extensive experiments on multiple vision tasks show that our proposed block brings improvements for different CNN architectures, and learns more diverse and complementary representations.

## 1   Introduction

The standard deep networks pass feature responses from lower-level layers to higher-level layers in a hierarchical fashion. With improved computational powers and novel network designs, stacking more layers has become a common and effective practice – the number of layers can be significantly large [9] and the connections between layers can be significantly dense [12]. Studies [29, 1] show that learned filters in the first few layers typically capture low-level texture in images, while the last few layers encode higher-level semantics. This structure is typically very effective for solving computer vision tasks, such as image classification [22], object detection [20, 9], semantic segmentation [8, 2] and video classification [14, 7, 24, 18].

Consider a conventional convolutional neural network, filters at each layer typically respond to the input response independently. As a result, redundant information could be accumulated across different channels in the same convolutional layer. As a pioneer, Squeeze-and-Excitation (SE) block [10] (Figure.1 (a)) is an architecture to explicitly model the channel-wise interaction, and demonstrates superiority on various vision tasks. However, SE block average each channel into a single scalar and modulate the responses through a simple multiplication. Therefore, it has a relatively limited bandwidth for the channel-wise communication.

In this paper, we introduce a network unit called Cross-channel Communication (C3) block, a simple yet effective operator that encourages information exchange across different channels at the same convolutional layer. In the C3 block, we first pass the response from each channel to a feature encoder, and then use a message passer to pass the information of one channel to all other channels. Then, we use a feature decoder to decode the message for each channel. The updated feature responses will then be passed to future layers and help perform down-stream tasks. We provide a high-level overview of the C3 block in Figure.1 (b).

The proposed module allows channels in each layer to communicate with each other before passing information to the next layer. Different from related network designs [10, 3, 6] where channels

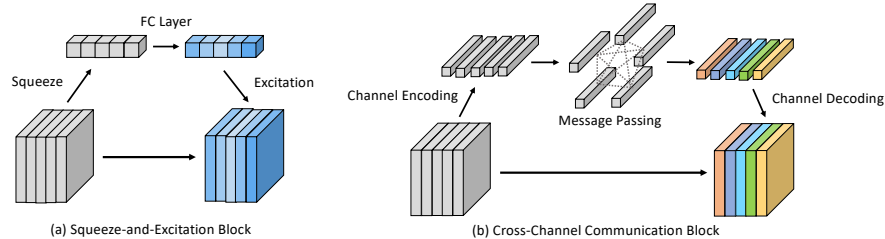

Figure 1: (a) network with Squeeze-and-Excitation (SE) block [10]; (b) our proposed cross-channel communication (C3) block. Without the squeezing operation in SE block, our C3 block enables a comprehensive communication across different channels at the same layer.

within each layer have limited interactions, our module enables channels to have a comprehensive interaction through a fully-connected graph structure. In C3 block, we retain the response map (no squeeze used) so that each channel has information of where *and* how the other channels respond to specific patterns in the image (e.g., different body parts of a person), and then introduce the feature encoder and decoder to enable thorough information exchange across channels, to learn diverse and complementary representations.

Experimental results demonstrate that the learned features are more effective for down-stream computer vision tasks such as image classification, semantic segmentation and object detection. To further validate our claim, we conduct experiments to analyze the behavior of C3 block. We find that 1) the correlations among channels in each layer are smaller than the baseline model, suggesting that filters in each layer can learn a more diverse set of representations; 2) when applying it to a shallower network, we can still achieve similar performance to deeper networks without C3 block, indicating that the learned features under C3 blocks are more representative.

## 2    Cross-channel Communication Unit

### 2.1    Formulation

In this section, we formally introduce the formulation of channel-wise interaction. Using 2D CNNs as an example, we illustrate this network module in Figure 2.

Let us consider a $L$-layer neural network architecture, where each layer has $n_l$ filters. In the $l$-th layer, the feature responses are denoted by $\boldsymbol{X}_l = \{\boldsymbol{x}_l^1, ..., \boldsymbol{x}_l^{n_l}\}$. In 2D CNNs, the response of each channel is a $H_l \times W_l$ feature map where $H_l$ and $W_l$ denotes spatial resolutions.

Generally, we can formulate the updated response after performing the channel-wise interaction by

$$\hat{\boldsymbol{x}}_l^i = \boldsymbol{x}_l^i + f_l^i(\boldsymbol{x}_l^1, ..., \boldsymbol{x}_l^{n_l}) \tag{1}$$

where $f_l^i$ is a function that takes all the feature responses at all channels, and updates the encoded features of a particular channel. We define *cross-channel communication* as the exchanged information across all channels, and the formulation of such information is modeled in $f_l^i$. In Squeeze-and-Excitation block [10], $f_l$ represents a squeeze and excitation operation [1]. In comparison, our proposed network structure enables a more comprehensive communication through a graph neural network by treating each channel as a node in the graph. We will discuss the details of the model in the following section.

### 2.2    Architecture

The proposed cross-channel communication network consists of three parts that are used for feature encoding, message passing and feature decoding, respectively.

**Feature encoding**. This module is used to extract the global information from each channel response map. Specifically, given the response map $\boldsymbol{x}_l^i$, we first flatten it to be an one-dimensional feature

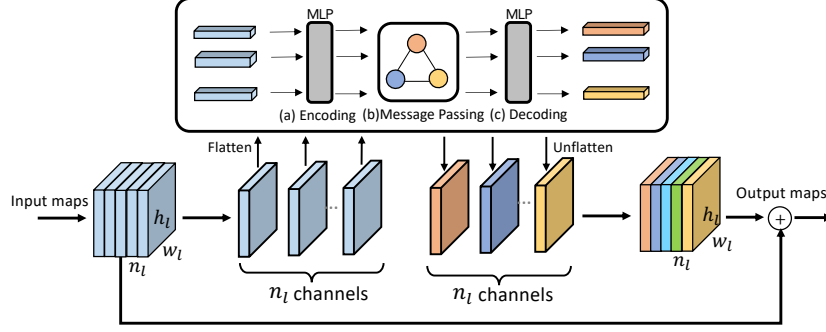

Figure 2: An overview of the Cross-channel Communication (C3) block. The feature responses in channels are passed to an encoder, and then the information is exchanged with other channels using a message passing mechanism. Finally, the features are decoded and added back to the input responses for the recalibration.

vector, and then pass it to two fully-connected layers:

$$\boldsymbol{y}_l^i = f_{enc}^{in}(\boldsymbol{x}_l^i), \quad \boldsymbol{z}_l^i = f_{enc}^{out}(\sigma(\boldsymbol{y}_l^i)) \tag{2}$$

In our model, $f_{enc}^{in}$ and $f_{enc}^{out}$ are two linear functions and $\sigma$ is a Rectified Linear Unit (ReLU).

In the feature encoding module, we add a bottleneck after $f_{enc}^{in}$ to reduce the feature dimension by a factor of $\alpha > 1$. This module compresses the features to reduce the computational cost. We set $\alpha = 8$ in our experiments.

**Message Passing**. This module enables channels to interact with each other, and update their feature responses so that it can encode a diverse set of representations of the input data.

The graph convolutional network [15] is a general framework for learning such interactions. Recently, graph attention networks [23, 27] have also been introduced, aiming to build a soft attention mechanism on top of GCNs. We use a similar formulation to model the cross-channel communication. Specifically, we construct a complete undirected graph, where $\boldsymbol{Z} = \{\boldsymbol{z}_l^i\}$ are nodes. We denote the edge strength between the two nodes to be $s_{ij} = f_{att}(\boldsymbol{z}_l^i, \boldsymbol{z}_l^j)$. Recent works use various methods to learn $f_{att}$ [24, 23], while we use a simple yet effective way to compute the edge strength:

$$\bar{z}_l^i = \sum_{k=1}^{h_l w_l} \boldsymbol{z}_l^i[k]/(h_l w_l), \quad s_{ij} = -(\bar{z}_l^i - \bar{z}_l^j)^2 \tag{3}$$

where $\boldsymbol{z}_l^i[k]$ is the $k$-th element of the flattened vector $\boldsymbol{z}_l^i$. In above, we use the average output from the feature encoder to increase the robustness in message passing period. We then compute the negative square distance to enable the channels with similar properties to have more communication, through which we want to group the similar channels and then make them diverse and complementary. We then feed them to a softmax layer to get the normalized attention scores $a_{ij}$, and then compute the output $\hat{\boldsymbol{Z}} = \{\hat{\boldsymbol{z}}_l^i\}$ as follows:

$$\hat{\boldsymbol{z}}_l^i = \sum_{j=1}^{n_l} a_{ji} \boldsymbol{z}_l^j \tag{4}$$

**Feature Decoding**. After acquiring the updated channel-wise outputs $\hat{\boldsymbol{Z}}$, the decoding module takes this information that contains the corrected beliefs of all channels, and reshape it to the same size of the input volume so that it can be passed to future layers using standard convolutional operations. We will add back this information to $\boldsymbol{X}_l$ as shown in Eq. (1).

The above mechanism ensures that channels at the same layer can communicate with each other comprehensively before passing information to future layers. The encoding layer captures the high-level information of each channel, the message massing module enables channels to interact with each other, and the decoding module collects the information and passes it to subsequent layers.

**Model and Computational Complexity**. Our module has a relatively low computational complexity. In each block, the computation only involves four FC layers. For the NC block at layer $l$, the additional parameters introduced is $N_l = \frac{4}{\alpha} \sum_{l=1}^{L} (H_l W_l)^2$, where $\alpha$ is the reduction ratio in our bottleneck layer as mentioned above. As a result, our module is independent of the number of channels, and thus introduces reasonable number of parameters for both small and large networks. In practice, we find it is not necessary to add cross-channel communication at all convolutional layers. Hence, the complexity is further limited by adding our C3 block to only a few separate layers.

## 2.3 Analyzing the Channel Responses

To understand how the communication affect the channel response maps, we compute the correlations among all the response maps within each layer, and compare the behavior with/without using NC block. Specifically, at each spatial location $[m, n]$ in the feature map, we compute the correlations:

$$c_l^{ij} = \frac{\sum\limits_{m=1}^{H_l} \sum\limits_{n=1}^{W_l} (x_l^i[m,n] - \bar{x}_l^i)(x_l^j[m,n] - \bar{x}_l^j)}{\sigma_{x_l^i} \sigma_{x_l^j}} \tag{5}$$

where $\bar{x}_l^i$ and $\sigma_{x_l^i}$ are the mean and standard derivation of $x_l^i$. We then take the absolute values of all the $c_l^{ij}$, and take the average. A larger value indicates that there is redundancy in the encoded features, while a smaller value means that the learned featuers are more diverse.

## 3 Related Works

Our design of the cross-channel communication network shares some high-level similarities with some recently proposed network units. We highlight a few most related works, and discuss their the differences. Broadly, we categorize these networks into two categories: modeling feature map interactions spatially or among each channel.

**Networks that Model Spatial Interactions.** There are network structures that learn spatial transformation to change input feature maps [13, 4]. Besides learning to perform spatial transformation, Wang *et al*. proposed a non-local network (NLN) to describe the inter-dependency in long-range spatial-temporal locations [24]. We also use graph neural network to model context within a single layer. However, NLN still models interactions for features spatially and primarily works for video data. In contrast, we model features within each channel. $z_l$ in Eq. (4) is updated using information from other channels, whereas each element of $z_l$ is updated using its own elements in NLN.

**Networks that Model Channel-Wise Interactions.** The Squeeze-and-Excitation Network [10] falls into this category in that it uses a simple network to calibrate feature responses. Besides performing a channel-wise scaling, [3, 6] proposed channel-wise attention for image captioning or semantic segmentation. In our formulation, the interactions across channels are modeled through a more comprehensive yet efficient mechanism. Concretely, in Eq. (4), SE block can be viewed as $\hat{z}_l = a_l z_l$. Similarly, the layer Normalization [17] network is yet another layer-wise operation, but with even simpler operations. More generally, various normalization methods such as group normalization [25] can be also regarded as a special case of channel-wise communication. However, the interactions across different channels are less powerful in that only a mean feature map and the standard deviation are learned.

## 4 Experiments

To demonstrate the effectiveness of the proposed module, we conduct experiments by plugging it into various network architectures to enable cross-channel communication within a layer. We mainly use the residual network [9] and its variants for our experiments. For clarity, we denote the union of all residual blocks with the same feature resolution as a *residual layer*. We first evaluate our model on image classification tasks, then verify its generalization ability to other tasks including semantic segmentation and object detection. Finally, we analyze the model behavior through ablation studies and visualizations.

## 4.1 Quantitative Comparison

**Image Classification**. We conduct experiments on two popular benchmarks: 1) CIFAR-100 [16], which has 100 object classes, and 500 images each for training and 100 for testing. (2) ImageNet [21], which has 1000 classes and more than 1.28M images for training, and 50K for validation.

We use representative network structures, such as ResNet [9], and Wide-ResNet [28]. We also compare our proposed module with the squeeze-and-excitation (SE) block [10]. For a trade-off between model complexity and performance, we add our C3 block to a few separate layers. Specifically, for both ResNet and Wide-ResNet, we add one C3 block to each residual layer by appending it at the front. For a fair comparison, we use publicly available code and the same training protocols (including data loader, learning rate, learning schedule, optimizer, weight decay, training duration) for all models. Specifically, we use stochastic gradient descent (SGD) with an initial learning rate 0.1, momentum 0.99, and weight decay $1e-4$ for both datasets. The learning rate is decayed by 10 after 100 and 140 epochs for CIFAR-100, and 30 and 60 for ImageNet. We report the average best accuracy of 5 runs.

Table 1 shows the classification errors on CIFAR-100 for different models and network architectures and the corresponding model size (in millions). Our proposed cross-channel communication module consistently outperforms the baseline and SE block [10] on various ResNet architectures. On Wide-ResNet, our network outperforms baseline model and is on par with SE network while introducing much fewer parameters (0.07M versus 0.40M). Note that on all these network architectures, our module introduces the same number of parameters because it is independent of the model size.

| | ResNet-20 | | | ResNet-56 | | | ResNet-110 | | | Wide-ResNet | | |
|---|---|---|---|---|---|---|---|---|---|---|---|---|
| | Size | FLOPs | Acc. | Size | FLOPs | Acc. | Size | FLOPs | Acc. | Size | FLOPs | Acc. |
| Baseline | 0.28 | 41.7 | 67.73 | 0.86 | 128.2 | 71.05 | 1.74 | 257.9 | 72.01 | 26.86 | 3.84G | 77.96 |
| Baseline + SE | 0.28 | 41.8 | 68.57 | 0.87 | 128.5 | 72.00 | 1.76 | 258.5 | 72.47 | 27.26 | 3.84G | **78.57** |
| Baseline + C3 | 0.35 | 46.0 | **69.34** | 0.93 | 132.5 | **72.27** | 1.81 | 262.2 | **73.36** | 26.93 | 3.87G | 78.34 |

Table 1: Classification accuracies (%) on CIFAR-100 [16] with different models.

In Table 2, we report the classification errors on ImageNet for different models along with the model size (in millions). We use standard ResNet-18, ResNet-50 and ResNet-101 as baselines for comparisons. We observe a consistent trend as in Table 1. Our cross-channel communication module outperforms baseline consistently, by introducing only 0.33M parameters. Meanwhile, our model achieves comparable performance to SE block, even though it introduces more parameters (over 3M parameters). In our experiments, we also observe that models with cross-channel communication modules consistently have lower errors in both training and validation over the whole training period.

| | ResNet-18 | | | ResNet-50 | | | ResNet-101 | | |
|---|---|---|---|---|---|---|---|---|---|
| | size | top-1 err. | top-5 err. | size | top-1 err. | top-5 err. | size | top-1 err. | top-5 err. |
| Baseline | 11.69 | 30.28 | 10.52 | 25.56 | 23.61 | 7.27 | 44.55 | 22.48 | 6.18 |
| Baseline + SE | 11.78 | 30.15 | 10.72 | 28.07 | **22.51** | **6.43** | 49.29 | 22.14 | 6.14 |
| Baseline + C3 | 12.02 | **29.30** | **10.48** | 25.89 | 23.19 | 6.60 | 44.88 | **21.93** | **6.02** |

Table 2: Classification errors on ImageNet [21] with different models.

**Object Detection and Semantic Segmentation** We use Faster R-CNN [20] for object detection on the COCO dataset [19], and Deeplab-V2 [2] for semantic segmentation on the Pascal VOC dataset [5]. We refer to [26] for the implementation of Faster R-CNN. We add the C3 block in those network structures, and report scores in Table 3. Specifically, for semantic segmentation, similar to the image classification task, we append one C3 block to each of the residual layers. For object detection, we only add one C3 block to the output of ROI pooling layer. We can see consistent improvements for two tasks. Recall that we only introduce a few additional parameters. Note that we train both models in a plug-and-play manner, which is different from the experimental settings in [11]. The goal of these experiments is to prove the generalization of our module in various tasks.

| Segmentation | Mean IOU | Mean Acc. |
|---|---|---|
| Deeplabv2 [2] | 75.2 | 85.3 |
| Deeplabv2 + SE | **75.6** | 85.6 |
| Deeplabv2 + C3 | **75.7** | **86.0** |

| Detection | Pascal VOC | COCO |
|---|---|---|
| Faster R-CNN [20] | 74.6 | 33.9 |
| Faster R-CNN + SE | 74.8 | 34.3 |
| Faster R-CNN + C3 | **75.6** | **34.8** |

Table 3: Performance on semantic segmentation on PASCAL-VOC-2012 (left) and object detection on PASCAL-VOC-2007 and COCO (right) with/without cross-channel communication). Bold indicates best results. For detection, mAP@(IOU=0.5) is reported for PASCAL-VOC-2007 and mAP@(IOU=0.5:0.95) is reported for COCO.

## 4.2 Analyzing the Communication Block

In this section, we systematically investigate the behavior of the C3 block from different aspects. Specifically, we will answer the following questions.

**Can we reduce the depth of the network when using C3 block?** Since channels at each layer can communicate and interact through our C3 block, one assumption is that a very deep network is not necessary to propagate information across channels. Therefore, we conduct experiments by adding only a few C3 blocks while reducing the depth of the neural network. We perform this ablated experiment on CIFAR-100 for image classification using ResNet [9] architecture with different number of residual blocks at each residual layer. In Figure 3, we see that using the C3 block, a shallower ResNet-74 can perform on par with ResNet-110 without C3 block, though with fewer parameters. This suggests that our C3 module can help to reduce the depth of neural networks while retaining the performance. As a reference, we further report the detailed number of parameters in these network structures in Table 4. As we can see, the networks with C3 block achieve better performance though having a similar amount of parameters to the baseline network. Through this ablation study, we demonstrate that the improvement of our model is not simply due to the increase in parameters.

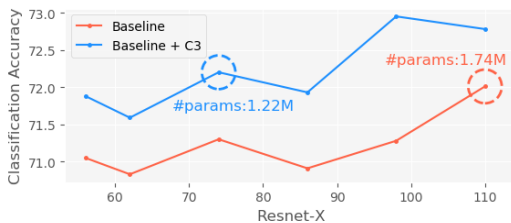

| | W/O C3 | W/ C3 |
|---|---|---|
| ResNet-56 | 0.86 | 0.93 |
| ResNet-62 | 0.96 | 1.03 |
| ResNet-74 | 1.15 | 1.22 |
| ResNet-86 | 1.35 | 1.41 |
| ResNet-98 | 1.54 | 1.61 |
| ResNet-110 | 1.74 | 1.80 |

Figure 3: The classification accuracy for ResNet with different number of layers.

Table 4: The corresponding model size for ResNet.

**Does C3 block reduces redundancy in features?** To understand the outcome of cross-channel communication, we compute the correlation scores (as described in Sec 2.3) among all the channel-wise features for the models without and with our C3 blocks. We track the correlation of channel responses during the whole training stage. As indicated in Fig. 4, after features are passed to the proposed module (Baseline + C3 After), the correlations among channels is consistently smaller (Baseline + C3 Before). When comparing to the baseline, both values are significantly smaller. This suggest that channels are effectively communicating with each other so that the encoded features become less redundant, and hence achieve a better performance.

**Is our model design helpful?** We investigate the extent to which the feature encoding/decoding and message passing contribute to the performance improvement. Specifically, we remove either the feature encoder/decoder or message passing from our C3 block, and perform image classification on CIFAR-100. As we can see in Table 5, both feature encoding/decoding and message passing improve the performance over the baseline network. These results demonstrate that both modules are necessary. The message passing helps channels to exchange information across channels so that each channel has a global information about the input. Without communication, the feature encoding and

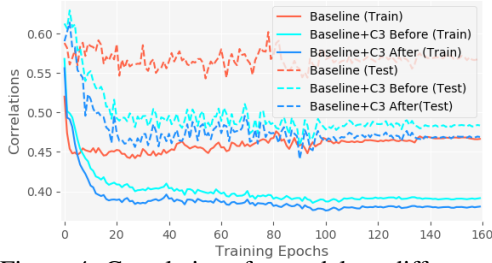

Figure 4: Correlations for models at different training stages.

| E-D | M-P | ResNet-20 | ResNet-56 | ResNet-110 |
|-----|-----|-----------|-----------|------------|
| - | - | 67.73 | 71.05 | 72.01 |
| ✓ | - | 68.70 | 71.95 | 72.65 |
| - | ✓ | 69.13 | 71.79 | 72.74 |
| ✓ | ✓ | **69.34** | **72.27** | **73.36** |

Table 5: Classification accuracy on CIFAR-100 for different C3 architectures. E-D is "Encoder and Decoder"; M-P is "Message Passing".

decoding help each channel to capture its own global structural information that can not be captured by a single or few convolution layers.

**Where should we add the C3 block in the network?** In CNNs, the lower-level layers typically encode low-level image features while higher-level layers contain semantic information [29]. In this experiment, we investigate the effect of adding the C3 block at different residual layers of ResNet. As indicated in Table 6, adding C3 blocks at the second or third residual layer is typically more effective. Note that due to larger feature size, the C3 block at first residual layer has more parameters than those in second and third residual layer. This further supports our previous claim that the improvement is not entirely due to the increase of model size. Instead, the C3 block at the second and third residual layer indeed learns more helpful information to help the task. An explanation is that filters at higher layer typically encode high-level semantic information [29], it's more likely get diverse and informative responses through communication. This indicates that we can further reduce the model size with few sacrifice of performance, by merely adding our C3 block in the last few layers.

| layer 1 | layer 2 | layer 3 | ResNet-20 | ResNet-56 | ResNet-110 |
|---------|---------|---------|-----------|-----------|------------|
| ✓ | - | - | 68.67 | 71.16 | 72.28 |
| - | ✓ | - | **69.53** | 71.88 | 72.78 |
| - | - | ✓ | 69.12 | 71.53 | 72.01 |
| - | ✓ | ✓ | 69.19 | 72.03 | 72.95 |
| ✓ | ✓ | ✓ | 69.34 | **72.27** | **73.36** |

Table 6: Classification accuracy for models with C3 block inserted at different residual layers.

### 4.3 Visualizations

We have shown in Fig. 4 that the proposed C3 block can help channels to learn more diverse representations. To further investigate what the C3 block has learned, we employ an off-the-shelf tool to visualize the class activation map (CAM) [30]. We use Resnet-101 and Resnet-101+C3 trained on ImageNet for comparison. As shown in first and second row of Fig. 5, the heat maps extracted from CAM for our model have more coverage on the object region, and less coverage on the background region. In the bottom row, for images which contain multiple objects, the filters learned from our model can localize all the objects, while the original model usually localize at most salient object region in the image. Furthermore, we show the top six mostly intense class activation maps from the last layer. This way we can directly check what each channel excites about and where they are. As shown in Fig. 6, the top activation maps from baseline model have more overlaps than the activation maps from the model with C3 block. These visualizations further demonstrate that C3 block can help the channels to learn more comprehensive and complementary filters.

## 5 Conclusion

In this paper, we introduced a novel network unit, called Cross-channel Communication (C3) block. Unlike standard hierarchical deep network architectures, we allow channels in each layer to communicate with each other. Through communication, channels at the same layer can capture global information and calibrate with other channels. To encourage the communication, we use a simple

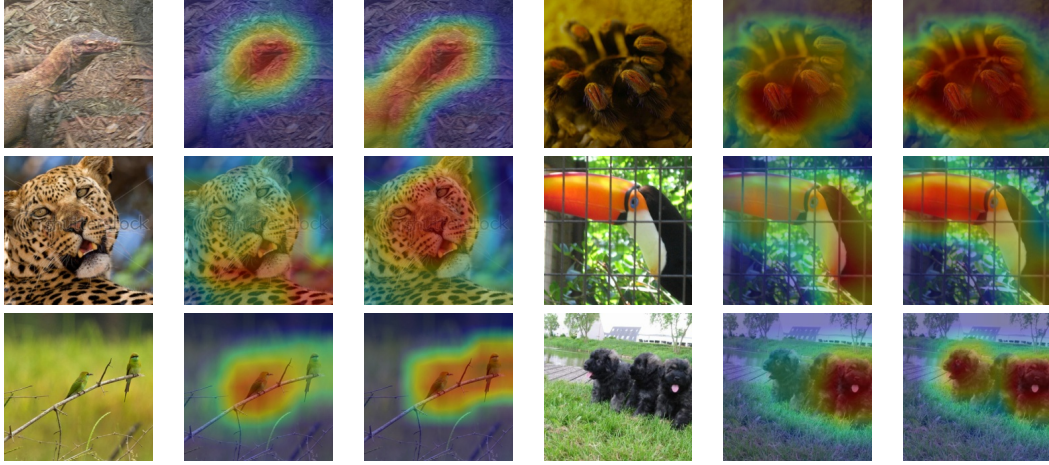

Figure 5: Class Activation Maps (CAM) at the last layer for ResNet-101 (2nd and 5th column) and ResNet-101 with C3 block (3rd and 6th column).

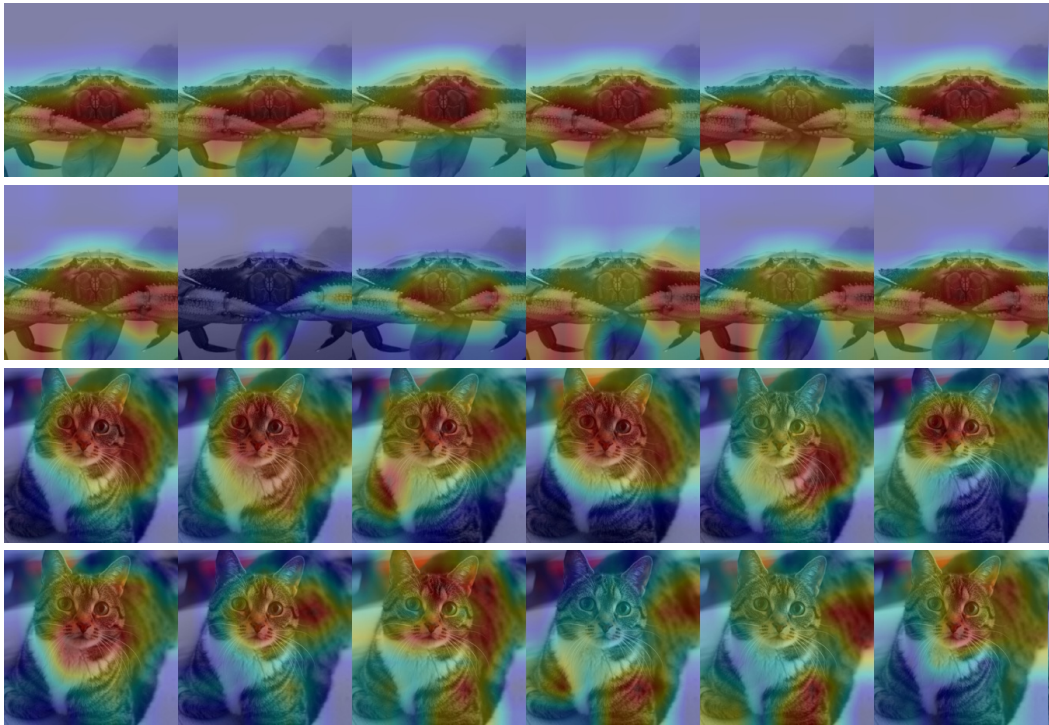

Figure 6: We visualize the top-6 mostly activated channels at the last layer. Odd rows are the class activation map from baseline ResNet-101; Even rows are from our model.

yet effective graph neural network that consists of a feature encoder, a message passing step, and a feature decoder. Our experimental results and ablation studies demonstrate that the C3 blocks can be added to modern network structures to advance the performance for a variety of computer vision tasks, with a small burden of model parameters.

**Acknowledgements.** The Georgia Tech effort was supported in part by NSF, AFRL, DARPA, ONR YIPs. The views and conclusions contained herein are those of the authors and should not be interpreted as necessarily representing the official policies or endorsements, either expressed or implied, of the U.S. Government, or any sponsor.

## Footnotes

[1]The original SE block does not have a residual addition, but we can reformulate it into a residual version by subtracting the scalar multiplier for each channel by 1

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
