[Supplementary Material]

# A    More Analysis on NC BLock

In this section, we further evaluate our NC block under different settings. We mainly focus on How the bottleneck affects the performance.

**How the bottleneck dimension affects the performance?** As we mentioned in our main paper, we used a bottleneck layer in our feature encoding and decoding stages, and the reduction ratio is 8. Here, we investigate how this reduction ratio affects the performance. We set the bottleneck ratio to $\{1, 2, 4, 8, 16\}$, and train ResNet-20 on CIFAR-100. As shown in Table 1, The NC block with reduction ratio $\alpha = 4, 8$ achieves better performance than other bottleneck ratios. This indicates that adding parameters into our NC block cannot guarantee the improvement of performance. Instead, adding a bottleneck to control the information flow is important in that it can promotes the neurons to communicate while retain their own information as well.

|          | 1     | 2     | 4     | 8      | 16    |
|----------|-------|-------|-------|--------|-------|
| Accuracy | 69.2  | 69.06 | 69.23 | **69.34** | 69.05 |

Table 1: Results on CIFAR-100 with different bottleneck dimension. The dimension reduction ratio $\alpha$ is chosen from $\{1, 2, 4, 8, 16\}$.

# B    Visualization

In this section, we employ an off-the-shelf tool to visualize the class activation map (CAM) [1]. In our main paper, we have shown that the proposed NC block can help neurons to learn more diverse and complementary representations, according to the calculated correlations. Here, we further visualize what the activation map looks like. We use ResNet101 and ResNet101+NC trained on ImageNet for comparison. Interestingly, comparing with the baseline model, we found two obvious differences:

- The neurons learned with our NC block are preciser. As shown in Fig. 1, it is apparent that the heat maps extracted from CAM for our model have more coverage on the object region, and less coverage on the background region. These results indicate that the neurons after communication, have learned more diverse and complementary information from the images for classification.

- The neurons learned with our NC block are more comprehensive. In Fig. 2, we show some images that contain more than one object of the same category. Obviously, the neurons learned from our model can localize all the objects in the images, while the original model usually localize on most salient object in the image. This comparison also demonstrate that our learned neurons are more diverse and comprehensive to capture all the objects in the image. Through communication, neurons know what the other neurons excite about and where they are. Hence, they can calibrate with each other to reduce the redundancy.

Furthermore, we dig more deeply on what each neuron has learned after adding our NC block. Specifically, we show the top 8 class activation map from the last layer separately. This way, we can directly check what each neuron excites on and where they are. As shown in Fig. 3, the activation maps from baseline model have more overlaps than the activation maps from the model with our NC block. These visualizations further demonstrate our neuron communication module can help the neurons to learn more comprehensive and complementary filters, through the interaction.

# References

[1] Bolei Zhou, Aditya Khosla, Agata Lapedriza, Aude Oliva, and Antonio Torralba. Learning deep features for discriminative localization. In *Proceedings of the IEEE conference on computer vision and pattern recognition*, pages 2921–2929, 2016.

Figure 1: We use CAM [1] to visualize the response map at the last layer. The first column are the input images, the middle column are the class activation map from baseline ResNet-101, and the last column are from our model.

| Input | CAM-Baseline | CAM-Ours |

Figure 2: The arrangement is same to Fig. 1. We can find the model with NC block can learn more comprehensive neurons that localize all objects in the images.

Figure 3: We use CAM [1] to visualize the response map of top-6 neurons at the last layer. Odd rows are the class activation map from baseline ResNet-101, and the even rows are from our model.