[Reviews · NeurIPS 2019]

Reviewer 1



originality:It is novel to encourage inter-neuron communication at the same layer. But it is similar to SE block or other attention moudle essentially. quality: the authors do many experiments clarity: easy to read significance:It is similar to SE block or other attention moudle essentially.

Reviewer 2



The authors propose an approach to increase the representation power of neural network by introducing communication between the neurons in the same layer. To this end a neural communication bloc is introduced. It first encodes the feature map of each neuron to reduce its dimensionality by a factor of 8. Then an attention-based GCN is used to propagate the information between the neurons via a fully-connected graph. In practice, a weighted sum of the neuron encodings is computed for each node, where the weights are determined by the nodes' features similarity. Finally, the updated representation is decoded to the original resolution and added to the original features. Importantly, this model applies the same operations to every neuron, thus the number of parameters is independent of the feature dimensionality, but dependent on the spatial size of the feature map. Several such blocks can be added at different layers of the network. In experiments on CIFAR and ImageNet the proposed module outperforms both the baseline and SE-Net using a variety of architectures. At the same time, it uses less parameters than SE-Net. That said, the absolute improvement over the baseline is within 1%. Additionally, improvements are demonstrated on the tasks of semantic segmentation and object detections. However, the method is only compared to the baseline here. The improvements are also within 1%. A qualitative analysis of the learned representations is provided. It shows that the NC module leads to learning neurons that are less correlated (more diverse). The paper is relatively well written and is easy to follow. Overall I find the idea to be reasonable and well presented. However, the proposed approach only achieves marginal improvements in practice, so I doubt it will have a significant impact on the community. Moreover, experiments evaluation is incomplete (see Improvements). The authors have partially addressed my concerns, however, I still find the improvements to be marginal (especially over SE-Nets), and the experimental evaluation to be incomplete (comparison to non-local networks is not reported). Overall, the paper is of a slightly above-average quality, but it will not have a significant impact on the community. I will increase my score to 6.

Reviewer 3



The paper is modestly original, high-quality in the experiments and analyses performed, written clearly enough (some improvements will be suggested below), and of modest significance. The NC block is an improvement over the SE block, and likely was developed with SE block as the main motivation. As such, a more thorough explanation of SE block decision and how NC block decisions contrast with it would be nice to include, and make the theoretical contribution clearer. The "algorithmic" novelty of the NC block feels modest, but could be improved with better explanation of the motivations behind its development and specific choices made. Experiments and analyses are sufficient and of high quality. What they show is that the NC block is usually but not always a modest improvement of the SE block. That does not feel very significant to practitioners.

[Author Response · NeurIPS 2019]

We thank the reviewers for the comments. In this work, we proposed a model to encourage the inter-neuron communication at the same layer (R1, R2, R3), and showed better performance than baselines and SE-Nets on image classification, semantic segmentation and object detection (R1, R2, R3). To demonstrate the effectiveness of our model, we did many (R1), high-quality (R3) experiments and presented reasonable (R2), qualitative (R2, R3) analysis on the learned models. All reviewers think the paper is clearly written and easy to read. We address reviewers' concerns below.

**(R1) Model Complexity vs. Performance**. As suggested, we report the complexity (FLOPs) for various models in Tables 1 and 2 (ResNet-56 has similar trend to other ResNets). Generally, for smaller networks (ResNets on CIFAR-100), our model has higher computational complexity than SE-Nets, while lower complexity for larger

|                  | ResNet-20 | | | ResNet-110 | | | Wide-ResNet | | |
|------------------|-----------|-------|-------|------------|--------|-------|-------------|-------|-------|
|                  | #Params.  | FLOPs | acc.  | #Params.   | FLOPs  | acc.  | #Params.    | FLOPs | acc.  |
| Baseline         | 0.28M     | 41.7M | 67.73 | 1.74M      | 257.9M | 72.01 | 26.9M       | 3.84G | 77.96 |
| Baseline + SE    | 0.28M     | 41.8M | 68.57 | 1.76M      | 258.5M | 72.47 | 27.3M       | 3.84G | **78.57** |
| Baseline + NC    | 0.35M     | 46.0M | **69.34** | 1.81M  | 262.2M | **73.36** | 26.9M   | 3.87G | 78.34 |
| Baseline + Convs | 0.39M     | 104.8M| 68.58 | 1.85M      | 321.0M | 71.57 | 36.1M       | 8.05G | 75.50 |

Table 1: Evaluating CIFAR-100 classification.

networks (Wide-ResNet on CIFAR-100 and all networks on ImageNet). We will include these statistics in the paper.

**(R1) It seems like more parameters achieve better results**. This is not true. In Table 2 of our submission, the networks with our NC blocks with fewer parameters can achieve better performance than those with SE blocks. This trend is also observed in our ablation studies: 1) In Table 6 of our submission, we show that putting the NC block at

|               | ResNext-50 | | | | MobileNet-v2 | | | |
|---------------|----------|-------|-------|------|----------|--------|-------|------|
|               | #Params. | FLOPs | top-1 | top-5 | #Params. | GFLOPs | top-1 | top-5 |
| Baseline      | 34.93M   | 5.89G | 23.85 | 7.12 | 3.50M    | 0.32G  | 28.12 | 9.71 |
| Baseline + SE | 37.45M   | 5.90G | 22.90 | 6.44 | 3.53M    | 0.32G  | 26.66 | **8.86** |
| Baseline + NC | 35.29M   | 5.89G | **22.51** | **6.23** | 3.51M | 0.32G | **26.29** | 9.09 |

Table 2: Evaluating ImageNet classification.

the second stage of ResNet is better than putting it at the first stage. Note that the NC block at first stage introduces more parameters due to a larger response map; 2) In Fig. 3 of our paper, we show that shallower networks with NC blocks, though have fewer parameters, outperform ResNets of larger size. In the bottom row of Table 1, the same architectures after replacing each NC block with two convolution layers that contain more parameters, perform less favorably. All these suggest that the improvement is not simply due to the increased model size.

**(R1, R2) Marginal Improvements**. We argue that with a very little increase in model size and complexity, a $1\%$ improvement on multiple tasks (classification, detection, and segmentation) is not marginal, especially over strong baselines like ResNets, faster R-CNN and Deeplab-v2. The relative improvements over SE-Nets in many cases are also promising.

| Model | Segmentation | | Detection | |
|-------|----------|----------|------------|------|
|       | Mean IOU | Mean Acc. | Pascal VOC | COCO |
| Baseline      | 75.2     | 85.3     | 74.6       | 33.9 |
| Baseline + SE | 75.6     | 85.6     | 74.8       | 34.3 |
| Baseline + NC | **75.7** | **86.0** | **75.6**   | **34.8** |

Table 3: Comparison with SE-Nets.

**(R2) Why different models are used on CIFAR-100 and ImageNet**. In our submission, we followed SE-Nets to report the performance for a different set of models on CIFAR-100 and ImageNet. For ImageNet, we selected representatie variants of ResNets. As suggested, we also report the performance on ImageNet for ResNext-50 and MobileNet-v2 in Table 2. As we can see, our NC block performs slightly better than SE-Nets for both models, which demonstrates the effectiveness of the proposed NC block.

**(R2) Comparison to SE Block on segmentation and detection**. Following previous work, we used ResNet-101 as the backbone for both segmentation and detection in our paper. For comparison, we report the performance for the baseline model with SE block in Table 3. It shows adding NC block consistently improves over the baseline and SE-Nets.

**(R3) Motivations of NC block design**. We were partially inspired by SE-Net and other channel-wise attention mechanisms [4,7]. SE block and channel-wise attention in [4] use a squeeze operation which ignores the spatial structure of channel response, and the scalar-based excitation operation further restrains the information flow across different channels. The channel-wise attention in [7] is similar to the message broadcasting in our NC block. Without the feature encoder and decoder, it reduces to summing up similar channels together. However, in NC block, we retain the response map (no squeeze used) so that each channel has knowledge of where *and* how the other channels respond to specific patterns in the image (e.g., different body parts of a person), and then introduce the feature encoder and decoder to enable thorough information exchange across channels, to learn diverse and complementary filters.

**(R3) Intuition behind Eq. 4**. First, we use the average output from the feature encoder to increase the robustness in message broadcasting period; Second, we compute the negative square distance to enable the channels with similar properties to have more communication, through which we want to group the similar channels and then make them diverse and complementary by adding the residuals predicted from the feature decoder.

**Miscellaneous. R1, R2**: We added SE blocks to baseline image classification networks following original paper exactly. For semantic segmentation and object detection, we add the SE blocks to the $4^{th}$ residual stage and fix the lower layers in ResNet-101, the same way we did for NC block. **R2**: The encoder and decoder in our NC block are both bottleneck architecture which contains two 1D convolution layers with feature dimension $d \rightarrow d/8 \rightarrow d$, and the second convolution layer is used to unsqueeze. **R3**: We will change our term "neuron communication" to cross-channel communication, and will polish the presentation of our model. We will also re-draw figure 1 and figure 2 in our paper.

[Meta-Review · NeurIPS 2019]

This paper introduces a novel approach for inter-neuron communication in the same layer in a standard neural network, a Neural Communication (NC) block. The NC block results in slightly better performance (absolute improvement of 1%) than previous similar methods (such as Squeeze-Excitation blocks) on image classification, semantic segmentation and object detection, and reviewers find the formulation to be more general and better motivated. The authors demonstrate that shallower networks with the NC block can achieve similar performance to deeper networks. They also provide a detailed and useful analysis of the properties of the learned representations, and show that the NC blocks leads to learning neurons that are less correlated. Reviewers are concerned that the improvement is only marginal, and that comparison to non-local networks is not reported. They suggest that the paper might be stronger if there is a better explanation of the motivations behind the NC block development and the specific choices made.